# Triboelectric Nanogenerators as Active Tactile Stimulators for Multifunctional Sensing and Artificial Synapses

**DOI:** 10.3390/s22030975

**Published:** 2022-01-27

**Authors:** Jianhua Zeng, Junqing Zhao, Chengxi Li, Youchao Qi, Guoxu Liu, Xianpeng Fu, Han Zhou, Chi Zhang

**Affiliations:** 1Center on Nanoenergy Research, School of Physical Science and Technology, Guangxi University, Nanning 530004, China; zengjianhuayu@yeah.net (J.Z.); 1911301064@st.gxu.edu.cn (H.Z.); 2CAS Center for Excellence in Nanoscience, Beijing Key Laboratory of Micro-Nano Energy and Sensor, Beijing Institute of Nanoenergy and Nanosystems, Chinese Academy of Sciences, Beijing 101400, China; zhaojunqing@binn.cas.cn (J.Z.); lichx28@mail2.sysu.edu.cn (C.L.); qiyouchao@binn.cas.cn (Y.Q.); liuguoxu@binn.cas.cn (G.L.); fuxianpeng@binn.cas.cn (X.F.); 3School of Nanoscience and Technology, University of Chinese Academy of Sciences, Beijing 100049, China; 4School of Materials Science and Engineering, Sun Yat-sen University, Guangzhou 510275, China

**Keywords:** triboelectric nanogenerators, active tactile stimulators, multifunctional sensing, artificial synapses

## Abstract

The wearable tactile sensors have attracted great attention in the fields of intelligent robots, healthcare monitors and human-machine interactions. To create active tactile sensors that can directly generate electrical signals in response to stimuli from the surrounding environment is of great significance. Triboelectric nanogenerators (TENGs) have the advantages of high sensitivity, fast response speed and low cost that can convert any type of mechanical motion in the surrounding environment into electrical signals, which provides an effective strategy to design the self-powered active tactile sensors. Here, an overview of the development in TENGs as tactile stimulators for multifunctional sensing and artificial synapses is systematically introduced. Firstly, the applications of TENGs as tactile stimulators in pressure, temperature, proximity sensing, and object recognition are introduced in detail. Then, the research progress of TENGs as tactile stimulators for artificial synapses is emphatically introduced, which is mainly reflected in the electrolyte-gate synaptic transistors, optoelectronic synaptic transistors, floating-gate synaptic transistors, reduced graphene oxides-based artificial synapse, and integrated circuit-based artificial synapse and nervous systems. Finally, the challenges of TENGs as tactile stimulators for multifunctional sensing and artificial synapses in practical applications are summarized, and the future development prospects are expected.

## 1. Introduction

As an important part of the intelligent perception system, tactile perception can provide object attributes and related information through physical contact, thus having the basic ability to perceive external physical stimuli. In recent years, the wearable tactile sensors, which major physical transduction mechanisms includes capacitance, piezoresistivity, and so on [1,2,3,4], have developed rapidly in the fields of intelligent robots, healthcare monitors, and human-machine interactions [5,6,7,8,9,10]. A great number of significant progresses in high performance has been achieved by coupling of functional materials and micro/nanoscale manufacturing technology. However, the power supply is still supplied by external power or batteries, which greatly limits the development in the direction of economy, portable and environmentally sustainable. With the demand of widespread distribution for tactile sensing devices in artificial intelligence, it is of great significance to create active tactile sensors that can directly generate electrical signals in response to stimuli from the surrounding environment.

In 2012, the Wang group first invented the triboelectric nanogenerator (TENG), which is a new energy conversion technology [11,12,13,14,15,16]. The fundamental theory of the TENG originates from Maxwell’s displacement current, which is caused by the time variation of the electric field plus a media polarization term. Moreover, the working principle of the TENG is based on the coupling contact electrification and electrostatic induction, that is, the triboelectric charges can be created on the surfaces of two materials in contact with different polarities, and further form a potential difference when the two materials are separated by an external mechanical force. The potential difference can drive free electrons to flow between two electrodes on the surface of the two materials established by an external load. In principle, all materials in nature have triboelectrification effect, so that any material with different charge affinity can be applied to build the TENG. According to different electrode configurations and/or different moving methods of the friction layers to realize the electrostatic induction process, four different working modes of the TENG have been developed, such as vertical contact-separation mode [17], single-electrode mode [18], lateral-sliding mode [19], and freestanding triboelectric-layer mode [20], which can be widely applied for the collection of various mechanical energy in the environment, including touching [21,22], sliding [23,24], rotation [25,26], impact [17,27], vibration [28,29], and so on. In addition to versatile working modes, the TENG has the advantages of flexibility in materials selection, simple structure, light weight, ease of fabrication and low costs [30,31]. Since being invented, the TENG has been developed rapidly in the four fields of micro/nano power sources [32], self-powered sensing [33], blue energy [34], and high-voltage power sources [35,36].

As one of the most important applications of TENG, a variety of active tactile sensors, which can directly convert the mechanical signal of the surrounding environment into electric signal, have been demonstrated for tactile perception [37,38,39,40]. Figure 1 shows the three development stages of TENGs as tactile stimulators from 2012 to 2021. In the initial stage, the tactile stimulators based on TENGs are mainly used for some single function sensing, such as pressure [41], trajectory motion [42], tactile imaging [43], and so on, which cannot satisfy the practical demands. After several years of research, the tactile stimulators have gradually developed into the integration of multi-sensory functions [44,45]. Various triboelectric tactile perception devices have been developed, which have demonstrated the sensing characteristic of TENGs in human-computer interactions. Besides, inspired by biological perception systems, the tactile stimulators are developing towards the direction of artificial sensory synapses to mimic more practical and complex nervous systems, such as artificial synapses and artificial neural [46,47]. After ten years of efforts, the active tactile stimulators based on TENGs have made great progress in sensing and artificial synapses.

Here, we focus on the research progress of TENGs as active tactile stimulators for multifunctional sensing and artificial synapses. First of all, the applications of TENGs as active tactile stimulators in pressure, temperature, proximity sensing, and object recognition are introduced in detail. After that, the research progress of TENGs as active tactile stimulators for artificial synapses is emphatically introduced, which is mainly reflected in the electrolyte-gate synaptic transistors, optoelectronic synaptic transistors, floating-gate synaptic transistors, reduced graphene oxides-based artificial neuromorphic synaptic, and integrated circuit-based artificial synapse and nervous systems. At the end of this review, we summarize the challenges and look forward to the future prospect of TENGs as active tactile stimulators for multifunctional sensing and artificial synapses in practical applications. By the research prospects for multifunctional sensing and artificial synapses, TENGs-based active tactile stimulators are expected for significant impact and potential applications in micro/nano-electro-mechanical systems (MEMS/NEMS), flexible electronics, robotics, artificial intelligence, and neuromorphic computation.

## 2. TENGs as Tactile Stimulators for Multifunctional Sensing

### 2.1. Pressure and Temperature Sensing

In the tactile sensing system, the pressure and temperature are the two most common external stimuli [52,53]. In recent years, many active tactile stimulators based on TENG for pressure and temperature sensing have been reported, see Table 1 [54,55,56,57,58]. Chen et al. [54] reported an active tactile stimulator based on TENG through convenient inorganic modification which can realize the pressure and temperature sensing, respectively. Figure 2a shows the schematic diagram of the tactile stimulator, which is a typical structure based on the single-electrode mode. The dielectric layer consists of the polydimethylsiloxane (PDMS) and the organo-montmorillonite-cetyltrimethylammonium bromide. The electrodes consist of four composite electrodes formed by silver nanowires (Ag NWs) and copper (Cu) films interconnected by a network pattern of zinc oxide nanowires (ZnO NWs). These intrinsically stretchable materials enable the device to have a tensile function of up to 580%. When the external pressure is in the range of 16~64 kPa, the short-circuit current of the tactile stimulator increases linearly versus pressure, as shown in Figure 2b. The sensitivity can be calculated as 0.04 μA/kPa by the method of linear fitting. Figure 2c shows that the current of the tactile stimulator also increases with the applied temperature. When the temperature increases from 19.4 °C to 34.9 °C, the current increases from 7.2 μA to 16.4 μA, and the sensitivity is calculated as 0.59 μA/°C. In addition, the texture, hardness, and position of the measured object can be identified with a location detection sensitivity of 2 mm from the output waveform of the device. Similarly, Song et al. [55] reported a self-healing electronic skin by utilizing a TENG with single electrode mode, which also has the ability to sense pressure and temperature due to the triboelectric-electrostatic induction effect and thermal sensitivity mechanism. Figure 2d shows the working principle of the device, which senses the pressure and temperature of the measured object by detecting the output current of the device. At a pressure range of 0.12~6.25 kPa, the device exhibits a high sensitivity of 3.55 nA/kPa. With the increase of pressure, the sensitivity decreases to 0.70 nA/kPa (Figure 2e). This is mainly because the accumulation of triboelectric charge is closely related to the contact area. When the pressure increases, the contact area gradually saturates and the accumulation of triboelectric charge also reaches the saturation state, which reduces the sensitivity. The device uses temperature-sensitive ionic liquid as an electrode, which the resistance will be affected by the temperature, resulting in a change in output current. Figure 2f shows the linear relationship between temperature and output current with a temperature sensitivity is 3.76 nA/°C. Although these studies show that the tactile stimulator can sense pressure and temperature separately through the output current as a detection signal, the signal is easily disturbed under the simultaneous action of the two stimuli, making it difficult to sense pressure and temperature simultaneously. Therefore, it is necessary to develop devices that can respond to different signals under different external stimuli. Rao et al. [56] reported a tactile electronic skin based on TENG with a single electrode mode, which can be used for simultaneously detecting temperature and pressure (Figure 2g). The output voltage of the device is used for sensing pressure, with a sensitivity of 5.07 mV/Pa in 200~1720 Pa, and a sensitivity of 1.89 mV/Pa in 1720~3650 Pa (Figure 2h). The output voltage is positively correlated to the surface triboelectric charge, which is directly proportional to the contact area. Therefore, the high sensitivity is primarily attributed to the pyramidal microstructure on the surface of PDMS. The composite materials of Bi_4_Ti_3_O_12_ and reduced graphene oxide are used as the thermoresistive electrode of the device to sense the temperature change through the electrode resistance as the detection signal. The detection range of temperature for the device is 25~100 °C, which has high sensitivity and is not affected by pressure (Figure 2i). Due to the large internal resistance of TENG, the electrode resistance value can change in a large range without affecting the output of the device. And the detection of pressure and temperature is two different signals that do not interfere with each other, the ability to sense pressure and temperature is realized simultaneously. This research provides a new strategy for wearable sensing, and a new concept in the fields of medicine, prosthetics, and robots, etc.

In addition to the above combination triboelectric and thermoresistive electrode, the polarization of pyroelectric materials changes as the temperature, which exhibits the advantage of self-powered. Ma et al. [57] designed a self-powered flexible tactile stimulator, which can distinguish pressure and temperature stimuli by the coupling mechanism of triboelectric–piezoelectric–pyroelectric multi-effect. Figure 3a shows the working principle of the device sensing pressure and temperature. The device is designed as a sandwich structure, which is composed of Ag NWs polytetrafluoroethylene (PTFE) film, graphene electrodes and polyvinylidene fluoride (PVDF) from top to bottom. The Ag NWs PTFE film act as the friction layer, which has excellent triboelectric property. The PVDF film has both piezoelectric and pyroelectric effects, which can be used as a temperature-sensitive layer, but also enhances the coupling with triboelectric potential. When the finger touches the Ag NWs PTFE film, compressive strain loaded on the PVDF causes the polarization density to increase, thereby forcing electrons to flow from the ground to the graphene electrode. After releasing the strain, the electrons will return to the ground. This process is the same as the trend of charge flow induced by triboelectric, which has a synergistic enhancement effect. Meanwhile, the PVDF can also respond to the temperature gradient of the tactile stimulator and the finger due to the pyroelectric effects. With the temperature rises, the polarization density of the PVDF reduces with the decreases of the electric dipole moment, so that electrons flow from the graphene electrode to the ground. Conversely, the electrons flow from the ground to the graphene electrode. This process is opposite to the trend of charge flow induced by pressure. Figure 3b shows the response of the device to pressure and temperature coupled stimuli, which continuously produces two different peaks in opposite directions. This is mainly due to the different response time of the device to pressure and temperature stimuli. Based on the principle of response time difference, the tactile sensor can simultaneously distinguish pressure and temperature stimuli without signal interaction. When the pressure is 10.5~52 kPa, the sensitivity is 0.092 V/kPa, and the minimum detectable pressure is 0.5 kPa (Figure 3c). For temperature sensing, the sensitivity is 0.11 V/°C in the range of 10~45 °C (Figure 3d). Although the tactile stimulator responds to pressure and temperature stimuli by using the output voltage signal, which can cleverly use the time difference of response to distinguish these two different external stimuli. In addition, the device also has the advantages of self-powered, multifunctional, and antibacterial, which is expected to be widely used in the fields of soft robotics, artificial prosthetics, and healthcare monitors. Similarly, Wang et al. [58] reported a multifunctional self-powered tactile stimulator based on the coupling mechanism of triboelectric–piezoresistive–thermoelectric multi-effect, which can achieve the sensing of pressure, temperature, and materials. The tactile stimulator consists of a multilayer structure superimposed with a PTFE film as the friction layer, a Cu sheet coated with Ag NWs as the electrode, and a spongy graphene/PDMS composite as the response element for piezoresistive and thermoelectric effects (Figure 3e). Based on the piezoresistive effect, the resistivity of the spongy graphene/PDMS composite changes with pressure. Therefore, it can respond to different pressure levels through the current measurement. The pressure sensitivity of the device is defined as *S* = (Δ*I*/*I*_0_)/Δ*P*, where *I*_0_ is the initial current when the pressure is released, Δ*I* is the current difference, Δ*P* is the pressure difference. The device has a high sensitivity of 15.22 kPa^−1^ for pressure sensing in the range of 0~5 kPa, and the sensitivity decreases as the pressure increases, as shown in Figure 3f. Based on the thermoelectric effect, the device can respond to temperature changes by detecting voltage signals. The voltage is defined as *V_therm_* = *S_T_*
*×* Δ*T*, where *S_T_* is the Seebeck coefficient, Δ*T* is the temperature gradient of the device. The temperature response range of the device is 25~60 °C, and the temperature coefficient is 35.2 μV/K. As the temperature gradient increases, the voltage signals gradually increase, as shown in Figure 3g. Based on the triboelectric effect, the voltage signals changes with the contact materials, so that the material properties can be inferred by analyzing the corresponding signals (Figure 3h). This research can simultaneously realize pressure and temperature sensing by integrating different mechanisms to generate different electrical signals, which provide a new approach for multifunctional sensors and have potential applications in wearable electronics and robotics.

### 2.2. Pressure and Proximity Sensing

Besides the above pressure and temperature sensing, the proximity sensing is also very important for tactile perception, which can detect the motion information of the object without contacting the measured object. Due to the coupling effect of triboelectrification and electrostatic induction, the TENGs as triboelectric skins can not only realize pressure sensing, but also achieve proximity sensing, the working mechanism of which is shown in Figure 4a [59]. The triboelectric skins use percolating silver-flake network as the extendable electrode, and rubber as the friction layer with pyramidal microstructure surface. In the initial state, the external object contacts with the friction layer, electrons transfer from the object surface to the friction layer surface owing to the different electronegativity. At this time, the positive and negative charges are balanced, and there is no electron flow in the external circuit. With the gradual separation of the object, free electrons flow from the embedded electrode to the ground, thereby generating the current until the maximum distance between the two is reached. When the object approaches again, free electrons flow from the ground to the embedded electrode, thereby generating the opposite current and until full contact to form a complete electricity generation cycle. In the pressure sensing mode, different pressure makes the contact area different, resulting in different open-circuit voltages. Meanwhile, it is closely related to the size of the pyramidal microstructure, and the triboelectric skins with a middle-sized pyramidal microstructure (H = 1.24 mm; W = 1.64 mm) show an ultra-high sensitivity of 0.29 kPa^−1^ under the low pressure (Figure 4b). The design of the pyramidal microstructure effectively expands the pressure sensing range and improves the sensitivity, which provides a good strategy for the improvement of device performance in the future. Moreover, the open-circuit voltage shows good stability in different strain cycles (Figure 4c). In the proximity sensing mode, different separation distances lead to variations in the electrostatic field on the friction layer surface, which corresponds to different open-circuit voltage (Figure 4d). This multifunctional active perception expands the application in the field of human-machine interactions. By coupling triboelectric and capacitance modes, Yuan et al. [60] reported a liquid–polymer tubular TENG that can realize pressure and proximity sensing in multiple modes. The friction layer of the device is a hollow tube made of silicone rubber, and the electrode is a mixture of conductive liquid and carbon nanotubes filled in the hollow tube, as shown in Figure 4e. Based on triboelectrification and electrostatic induction, the device can be used as a typical TENG of the single electrode mode. In the low force range, the device shows a high sensitivity of 43.91 V/N (Figure 4f). As the force increases to approximately 10 N, the sensitivity value drops to 3.029 V/N. When the two tubes are stacked together, the cross area is equivalent to a capacitor, which can also realize pressure and proximity sensing. Although the maximum sensitivity of the force in capacitance mode is only 18.81 N^−1^, the maximum distance of proximity sensing can reach 400 mm (Figure 4g). Furthermore, in order to further realize the proximity sensing of the robotic hand, a stretchable sensor array with two modes of TENG and capacitance is developed, as shown in Figure 4h [61]. The sensor array uses a cross-grid liquid metal layer as electrode, and silicone rubber as dielectric layer and packaging material, which leads to the whole device being highly stretchable. This cross-grid structure enables the device simultaneously to realize TENG mode and capacitance mode. Under the action of TENG mode, the device can identify the position and pressure of the measured object by detecting the output voltage of the device and shows a high sensitivity of 0.68 V/kPa under the low-pressure range (<3.5 kPa). Under the action of capacitance mode, the proximity sensing and pressure sensing can be realized at the same time by monitoring the capacitance change of the device. The geometry design of electrode and the microstructure design of dielectric layer can improve the proximity sensing performance and pressure sensing sensitivity, respectively. Meanwhile, the hollow micro column microstructure designed in the dielectric layer can distinguish pressure and tensile strain stimuli (Figure 4i). This is because the thickness of the dielectric layer becomes smaller after stretching, which leads to increased electrostatic induction. Moreover, the device still shows excellent reliability under multiple cycles with tensile strain of 30%. Based on this design idea, the device can also be applied to proximity sensing and tactile perception of other parts of the robot.

### 2.3. Pressure and Object Recognition

In order to expand the sensing application of the tactile stimulators in a complex and diversified environment, it is very important to distinguish the material, hardness, shape, and size of the measured object for object recognition. Since different materials in nature have different abilities to gain or lose electrons, the tactile stimulators based on TENGs are particularly suitable for material recognition. Wang et al. [43] systematically investigated the influence of the contact pressure and contact materials on the output characteristics of the device. The device is a triboelectric sensor matrix of the single-electrode mode, which has the advantages of compact structure, small size, high sensitivity, and high resolution, as shown in Figure 5a. When the pressure is less than 80 kPa, the device shows a nearly linear response with a high sensitivity of 0.06 kPa^−1^ (Figure 5b). Moreover, with increasing pressure, the output voltage of the device gradually saturates. This is mainly due to the fact that at low pressures, the contact area gradually increases with increasing pressure. When the pressure exceeds 80 kPa, the contact area gradually becomes saturated, thus reducing the sensitivity. By contacting with several commonly used materials, the corresponding output voltages of the device are different, so that the material properties can be identified by analyzing the output voltage signals (Figure 5c). Moreover, the device still maintains stable output characteristics under more than 10,000 cyclic contact testing. This work effectively validates that the TENGs-based tactile stimulators can realize pressure and material recognition. Furthermore, Li et al. [62] reported a tactile stimulator integrated by multiple TENGs, which can realize pressure sensing, material, and hardness recognition. The tactile stimulator is composed of a traditional contact-separation mode TENG and a single-electrode mode TENG, as shown in Figure 5d. Based on triboelectrification and electrostatic induction, pressure sensing can be realized by analyzing the output voltage signal of the traditional contact-separation mode TENG. When the pressure is in the range of 40~140 N, the device shows a high sensitivity of 0.028 V/N (Figure 5e). The device can also realize material and hardness recognition by analyzing the output current of a single-electrode mode TENG, in which the different current values can distinguish the material of the measured object, and the change process of the current can recognize the hardness of the measured object (Figure 5f). This is because when the device contacts with soft materials, the deformation process of the contact surface is uniform and continuous, so that the amount of charge gradually accumulates, and the output current gradually reaches the peak. When contacting with hard materials, the output current of the device suddenly increases to the peak. Therefore, pressure sensing, material and hardness identification can be quantified by the output voltage, peak value, and duration time of the output current, respectively. This study shows that multimode tactile stimulator has broad application prospects for the sensing and detection in the field of intelligent robots.

In addition to the object recognition mentioned above, the gesture recognition of the robot hand during the grasping of the object is also highly important. Zhang et al. [48] reported a stretchable triboelectric-photonic smart skin, which can realize the pressure sensing and gesture recognition for a robotic hand. The smart skin is a single-electrode mode TENG, in which the stretchable Ag NWs are used as the electrode, the silicone rubber of the package device is used as the friction layer, as shown in Figure 5g. The aggregation-induced emission powder is mixed into the silicone rubber, which is tightly bonded to microcracked metal film. Generally, the microcracks of the metal film are detrimental to electrical properties of the electronics due to the reduction of electrical conductivity. However, the microcracked metal film can form a grating structure under tensile action, which can further regulate the photoluminescence intensity of the stretchable substrate. Meanwhile, electrical signal in response to physical contact with the object being measured, thus it can be used for pressure sensing without a power source. The pressure sensing characteristic can remain stable under different tensile strain levels, and has a high sensitivity of 34 mV/Pa in low pressure range. By attaching the smart skin to the robotic hand, the photoluminescence intensity and electrical signal can be detected to realize pressure sensing and gesture recognition, and there is no signal interference between each other (Figure 5h). Therefore, the device coupling triboelectrification and tunable photoluminescence realizes multifunctional sensing, which is expected to have important applications in the fields of flexible electronics, human-machine interactions, and intelligence robots. The tactile stimulator based on TENG typically uses soft materials, which makes it more compatible with soft robotics. Figure 5i shows a soft-robotic sensory gripper system, which is composed of a soft gripper and two tactile sensors based on TENG [63]. Among them, the length TENG sensor similar to the gear structure can measure the bending angle of the soft gripper, that is, the voltage signal generated by the device is detected. Secondly, the tactile TENG sensor with the patterned electrode can test sliding, contact position, contact area of the object through the triboelectric signal generated by the grating electrode and the object. Furthermore, the system can successfully realize the functions of the gripping status perception and object recognition by using machine learning technology for data analysis. This real-time signal processing ability also shows that the system has enormous potential in the field of human-machine interactions. Moreover, the digital twin application demonstration is successfully created for the management application of virtual assembly lines and unmanned warehouses (Figure 5j). The application program can realize object recognition and repetitive robot operation in virtual environment through the real-time operation of the system. This work cleverly integrates two different TENGs into soft-robotic gripper, and combines machine learning to achieve object grasping and recognition, which has broad application prospects in the fields of unmanned warehouses, smart factories, intelligent assembly lines, and the Internet of things.

## 3. TENGs as Tactile Stimulators for Artificial Synapses

### 3.1. Transistor-Based Artificial Synapses

#### 3.1.1. Electrolyte-Gate Synaptic Transistors

Electrolyte-gate transistors (EGTs) can availably tune the channel conductivity of the transistors by utilize ions in the dielectric layer, which shows a huge application prospect in the implementation of synaptic devices [64,65]. For generality, electrolytes are not just ionic conductors, but also insulators of electrons and holes. Electrostatic modulation and electrochemical doping are the two main principle of EGTs [66,67,68]. The first one is that the functional ions can accumulate at the interface between the semiconductor and the dielectric under the action of the external gate electric field, thus forming a double electric layer (EDL) with high capacitance. This type is a reversible process, so it has potential applications in emulating short-term synaptic plasticity. Secondly, as the external gate electric field increases, the functional ions are permeable, and the conductivity of the channel can be further doped and modulated in a nonvolatile manner. This type is an irreversible process, therefore it has potential applications in emulating long-term synaptic plasticity. The EGFs are widely used to mimic synaptic functions, owing to the migration of the functional ions under the action of the external gate electric field is very similar to the transfer of neurotransmitters [69]. In recent years, many studies have reported that the triboelectric potential of TENGs can replace the gate voltage of the electrolyte gated synaptic transistor to better mimic synaptic functions. For instance, Liu et al. [49] reported a self-powered synapse transistor (SPST), which is composed of a TENG and an electric-double-layer organic field effect transistor (EDLT), as shown in Figure 6a. The TENG can not only provide voltage for the gate of the EDLT to significantly reduce the power consumption of the SPST, but also can serve as a tactile stimulator to significantly enhance the active interaction of the SPST due to the external pressure is directly related to the output signal. In the EDLT, the ionic liquid LiTFSI is used as electrolyte and the PDVT-10 as semiconducting layer, respectively. By touching the TENG, due to the coupling effect of triboelectrification and electrostatic induction, electrons on the top Cu electrode of the TENG flow to the Si gate of the EDLT, thus the EDLT generates a negative gate voltage of and further forms the EDL. Under the action of the EDL, the channel carrier density of the SPST increases, so that the output signal is increased. As the touch is released, electrons flow from the Si gate to the top Cu electrode of the TENG, and the EDL effect gradually disappears, resulting in a decrease of the output signal of the SPST. According to this principle, the SPST can mimic the typical synaptic functions of biological (Figure 6b), including excitatory postsynaptic current (EPSC), paired-pulse facilitation (PPF), short-term plasticity (STP), and long-term plasticity (LTP). Furthermore, under the action of multiple TENGs, the SPST can combine multiple input signals and realize logic functions, which is helpful in the development of neuromorphic computation in the future.

Similarly, Figure 6c shows a self-powered artificial sensory memory system, which is composed of a triboelectric sensory receptor (TESR) and an electric double layer field effect synaptic transistor (FEST) [70]. The triboelectric layer of the TESR adopts a microstructure design, which makes the system has the characteristics of rapid-response and high sensitivity. In the FEST, the ionic liquid LiTFSI is used as an electrolyte, and another organic semiconductor material DPP-DTT as a semiconducting layer. Under different external pressure stimuli, the system successfully mimics the typical synaptic functions, such as EPSC, PPF, sensory memory (SM), short-term memory (STM), and long-term memory (LTM). The system actuated by the TESR can directly enhance the interaction between the environment and devices, which has widespread application prospect in the field of human-machine interactions (Figure 6d). Furthermore, Zhang et al. [71] reported an energy-efficient and easy-to-fabricate tactile-sensing element (TSE) by combining a TENG and an EGT, which can also emulate the typical synaptic functions. Besides, the TSE the device can simulate the ability of spiders to recognize prey by sensing the vibration of spider webs.

In addition to the aforementioned semiconductor material, two-dimensional semiconductor materials have also been used as channel materials for EGTs [72,73,74,75,76,77,78,79,80]. Among them, MoS_2_ has remarkable charge carrier mobility and tunable bandgap, which has attracted special attention of researchers [81,82,83,84]. Figure 7a shows a versatile MoS_2_ transistor actuated by sliding-mode TENG, which can modulate its electrical properties through triboelectric potential [85]. With the change of the relative displacement, the output characteristics of the TENG shows a stable periodic change, as shown in Figure 7b. The change process can be divided into four states. (i) when the relative displacement of the TENG is D+, the output voltage gradually increases from 0 to 2.15 V, and a positive triboelectric potential is formed at the MoS_2_ transistor gate. (ii) As the relative displacement of the TENG returns to the initial position, the output voltage returns from 2.15 to 0 V. (iii) Conversely, when the relative displacement of the TENG is D−, the output voltage gradually decreases from 0 to −1.36 V, and a negative triboelectric potential is formed at the MoS_2_ transistor gate. (iv) As the relative displacement of the TENG returns to the initial position again, the output voltage returns from −1.36 to 0 V. Under the actuating action of the TENG, the MoS_2_ transistor has three working states: flat-band state, accumulation state, and depletion state. The MoS_2_ transistor uses a proton conductor as an electrolyte, which can quickly migrate/polarize under the action of electric field and form a high capacitance EDL compared with other ions with larger size and volume [86,87]. In an analogy sensory neuron system, the TENG can be used as a tactile stimulator to induce the presynaptic signals, and the MoS_2_ transistor can be served as a postsynaptic device to output corresponding postsynaptic current (PSC), as shown in Figure 7c. Under the action of a displacement pulse, the PSC increases with the increase of the displacement distance (Figure 7d). Under the action of two continuous displacements, the PSC peak induced by the second displacement is greater than that induced by the first displacement (Figure 7e). The output characteristics of the MoS_2_ transistors are highly similar to the synaptic behavior of biological. Although TENG as a tactile stimulator can effectively reduce the power consumption by replacing the gate voltage of the synaptic device, there is still a large gap compared with the consumption of only 1~100 fJ each synaptic event in the brain. Therefore, it is still necessary to further reduce the power consumption of the synaptic device, which is of great significance to the development of the synaptic device. Sun et al. [51] proposed a triboelectrification-activated artificial afferent neuron at femtojoule energy (Figure 7f). The triboelectrification activation can induce the migration of anion/cation and form the EDL in the ion gel, which can effectively transfer the Fermi level of MoS_2_ channel to achieve synaptic plasticity. The migration process of the anion/cation in the EGT is crucial for mimicking the working process of biological synapses. The source-drain tunneling current can be effectively reduced and the transport properties in the channel can be effectively improved due to the atomic thickness and wide band gap of the MoS_2_ [83]. Therefore, the MoS_2_ channel still maintains excellent output characteristics and synaptic plasticity when the drain voltage drops to 1 mV, and the power consumption of this system can be greatly decreased to 11.9  fJ per spike (Figure 7g). The above-mentioned studies show that the EGTs actuated by TENGs as tactile stimulators have broad application prospects in the fields of low power bionic electronic devices and artificial neural networks in the future.

#### 3.1.2. Optoelectronic Synaptic Transistors

As an important part of optoelectronic integrated artificial neural network, optoelectronic synaptic transistor can directly convert presynaptic light stimulation into postsynaptic current signal, thus mimicking the function of visual nervous system [88,89,90]. In order to mimic a more practical and complex nervous system, it has been reported that the triboelectric potential as the gate voltage of the optoelectronic synaptic transistor can cooperate with mechanical and optical stimulation to achieve the multimodal plasticity. Figure 8a shows a mechano-photonic synapse device that can synergize mechanical and optical stimulation, which is composed of a TENG and a graphene/MoS_2_ heterostructure transistor [91]. The integrated TENG uses a vertical contact-separation mode, in which the Cu electrode acts as a movable friction layer, and the PTFE/Cu friction layer is integrated on the transistor to provide different gate voltages by changing the mechanical displacement. For the graphene/MoS_2_ heterostructure transistor, the photogenerated carriers can be transferred from MoS_2_ to graphene due to the poor intrinsic optical responsivity of graphene. The interfacial barrier impedes the recombination time of photogenerated electron-hole pairs compared to the carrier transport time through graphene, resulting in sustained photoconductivity, which is more conducive to mimicking the decay behavior of biological synapses [92]. When the device is in the dark state, the initial PSC increases significantly from 260 to 308 μA with the increase of displacement (Figure 8b). With the increase of light intensity, more photogenerated carriers are activated, which leads to the decrease of graphene conductivity and PSC. When the displacement is fixed to 1 mm, the PSC increment enhances from 26 to 35 μA as light intensity (green light pulse; pulse width, 50 ms) enhances from 0.73 to 13.5 mW/cm^2^ (Figure 8c). As the light was turned off, the current first decreased slightly and then remained stable, showing repeatable LTD synaptic behaviors. In addition to the simulation of synaptic functions, the recognition accuracy of artificial neural network simulation can be further evaluated by increasing the number of synapses and training samples. When the number of synapses is 60,000, the recognition rate is 92% (Figure 8d). The synapse device is effectively synergistic the interaction of mechanical and optical to achieve dual-modal synaptic plasticity, which makes them have huge application prospects in the fields of interactive neuromorphic computation, human-machine interactions, and artificial intelligence, and so on.

Moreover, it has been recently reported that optoelectronic synaptic transistor based on perovskite materials can also be used for the synaptic plasticity due to the excellent light absorption of perovskite materials. Figure 9a shows a multisensory integration nervous system integrated by flexible TENG and the photosynaptic transistor with perovskite (Cs_2_AgBiBr_6_) quantum dots, which can simultaneously realize haptic and iconic perception recognition [93]. The flexible TENG can be used as skin receptor, while the quantum dots act as retina receptors, which can convert haptic and optical stimuli into presynaptic electrical signals (pre-synaptic neurons 1 and 2). Then the channel current of the photosynaptic transistor is obtained by the tuning of the bottom gate (pre-synaptic neuron 1) and floating gate (pre-synaptic neuron 2) of the photosynaptic transistor. Under the action of a single tactile stimuli, the triboelectric potential generated by the TENG can induce the photosynaptic transistor to form a gate electric field, which makes the quantum dots able capture charge carriers in the semiconductor and play a role in tuning the channel current. With the disappearance of the electric field, the charge carriers can stay on the quantum dots floating gate for a certain time, which makes the photosynaptic transistor display synaptic characteristics of PPF and LTM. When the time interval between two successive tactile stimuli enhances from 20 ms to 40 s, the PPF index reduces from 168% to 102%. Moreover, under the action of 30 continuously touch pulses, the channel current increases with the enhance of light intensity (Figure 9b). Under the action of a single optical stimuli, the photoinduced excitons generated by quantum dots can be quickly separated to construct built-in electric field so that the quantum dots are able to capture the charge carriers. By the same principle, when the time interval between two successive optical pulses enhances from 20 ms to 20 s, the PPF index reduces from 136% to 102%. Meanwhile, the channel current can be adjusted by different pressure under 30 continuously optical pluses (Figure 9c). Further, the recognition accuracy of the multilayer perceptron neural network is 68.02%, 80.55% and 86.83% under single iconic sense, haptic sense, and multisensory integration after 130 training epochs, respectively (Figure 9d). These results indicate that multisensory integration effectively improves the recognition accuracy of the external environment. This work shows that the integration of TENG and photosynaptic transistor is of great significance for the development of interactive neuromorphic computation.

#### 3.1.3. Floating-Gate Synaptic Transistors

In addition to the above synapse transistors, a tribotronic floating-gate neuromorphic transistor has been proposed by combining the silicon-based field effect transistor and integrated TENG, which can also realize multiple synaptic plasticity by external stimulation. Figure 10a [94] illustrates the schematic diagram of the tribotronic floating-gate neuromorphic transistor. There are TENG parts on the bottom of the transistor, and the friction layer composed of the Cu electrode and fluorinated ethylene propylene (FEP) can continuously contact and separate with the Cu film at the bottom of the transistor under the action of external force. The gate electrode and the dielectric layer of the transistor are composed of heavily doped Si and 300 nm thick SiO_2_, respectively. Au nanoparticles deposited on SiO_2_ dielectrics are used as floating gate layer of the transistor, which synergistically functionalizes with the 10 nm thick HfO_2_ tunneling dielectric layer to achieve the charge trapping/detrapping process [95]. The MoS_2_ transferred on HfO_2_ is used as the channel of the transistor. The Au nanoparticles can obtain the induced charge carriers under the interaction of the triboelectric potential and the MoS_2_ channel, and progressively decay back under the action of the tunnel layer. This process can effectively simulate the transfer of neurotransmitters, and more sophisticated mechanoplasticized synaptic behaviors can be achieved through the synergy of tribotronic control gate and charge-trapping floating gate, which is completely different from the electrolyte gated synaptic transistors discussed above. Under a displacement action, the PSC of the transistor shows four different stages (Figure 10b). The displacement distance is 0.55 mm, and the time is 0.1 s. In stage I, the PSC always keeps at about 1.2 μA due to the displacement is not performed. With the start of the displacement, the PSC instantaneously enhances to 2.8 μA (stage II) and then decreases rapidly (stage III). This is because the induced electrons in the MoS_2_ channel are partially obtained by the Au nanoparticles, and the obtained negative charges continue to affect MoS_2_ channel with the release of displacement, thus inducing inhibitory postsynaptic current (IPSC). In stage IV, the PSC exhibits a typical LTP behavior due to the obtained charges by the Au nanoparticles gradually tunnel back. In order to better understand the physical mechanism of the synaptic device, Figure 10c shows the charge transfer process of each stage and corresponding energy band state. In stage I, the positive and negative charges are in balanced due to the contact between FEP and Cu film, so that the Fermi level of the MoS_2_ channel does not change. With the gradual separation of FEP and Cu film, the positive charge in Cu film induces the synaptic transistor to produce a positive gate voltage, which causes the accumulation of electrons in the MoS_2_ channel and the downward bending of the energy band. Simultaneously, the Au nanoparticles floating gate can capture some induced electrons in MoS_2_ channels. When the FEP contacts Cu film again, the positive charge in Cu film returns to equilibrium again, thus the induced gate voltage disappears. However, the electrons captured by the Au nanoparticles cannot return immediately and form a negative gate voltage, which causes the electrons depletion in the MoS_2_ channel and the upward bending of the energy band. During this time, the synaptic transistor shows the synaptic characteristics of IPSC. As the electrons captured by the Au nanoparticles gradually return to the HfO_2_ tunneling dielectric layer, the negative gate voltage gradually decreases, so that the electrons depletion in the MoS_2_ channel and the upward bending of the energy band gradually weaken. During this process, the synaptic transistor shows the synaptic characteristics of LTP. The neuromorphic transistor successfully mimics typical synaptic plasticity behaviors such as potentiation/inhibition and paired pulse depression/facilitation (Figure 10d,e), which has broad application prospects in neuromorphic computation, learning and memory behavior simulation, etc.

### 3.2. Reduced Graphene Oxides-Based Artificial Synapse

As a derivative of graphene, reduced graphene oxides (rGOs) have the characteristics of controllable defects and adjustable functional groups, which can effectively capture external electrons with nonvolatile electron-trapping characteristics. Wu et al. [50] reported an artificial synapse device based on rGOs, which can realize the typical functions of the neuromorphic system (Figure 11a). The key component of the device uses the hybrid layer of polyimide (PI) and rGOs as the friction layer, and the aluminum (Al) foil electrode is attached to the PI film substrate, the schematic structure is shown in Figure 11b. The rGOs sheets can act as electron traps to effectively promote the charge transfer from PI to rGOs sheets, which play a key role in mimicking the typical functions of the nervous system [96]. When the Al film on the top contacts the friction layer for the first time (i.e., the first training), the electrons are transferred from the Al film to the surface traps in the friction layer (Figure 11c). As the Al film gradually separates from the friction layer, most of the electrons in the surface traps are then transferred to the body traps. In the second training, the electrodes have the same transfer process, which increases the total number of electrons in the body trap. After sufficient training, the number of electrons captured in the body trap reaches the maximum value, which gradually saturates the output voltage of the synaptic device (Figure 11d). From the perspective of bionics, this process is similar to biological excitatory postsynaptic potential. The output voltage not only contains the current press information but also accumulates the historical information of previous stimulations, which can also be treated as the learning behavior of humans. In addition, the stacking structure in the friction layer can effectively regulate the duration time of the information received by the synaptic device, thereby realizing the advanced synaptic behaviors of STM and LTM. This work is significantly different from the above reported synaptic transistor devices. The typical functions of the neuromorphic system can be realized by using the materials with nonvolatile electron-trapping characteristics as the friction layer of TENG, which greatly simplifies the structure of synaptic device and plays an important role in promoting the development of synaptic devices in the future.

### 3.3. Integrated Circuit-Based Artificial Synapse and Nervous System

Besides the above synapse devices, TENGs-based tactile stimulators can also support dedicated integrated circuits to achieve synaptic functions. Figure 12a [97] shows a hardware-based biomimetic artificial slowly adapting type I tactile peripheral nervous system, which has the characteristics of ultra-high sensitivity and low power neuromorphic computation. Inspired by the structural characteristics of biological tactile peripheral nervous system, the artificial system uses the components of TENG and transistor as slowly adapting type I mechanoreceptor, while the integrated circuit is used to simulate spike initiation and synaptic function. The power consumption of the artificial system in the inactive state is less than 15 nW and 28.6 mW in the active state, which effectively overcomes the disadvantage of electronic skin that does not expand easily on a large scale under the traditional architecture, that is, the tactile signal transmission delay and power consumption do not enhance significantly with the increase in the number of mechanoreceptors and artificial neurons. Figure 12b shows a two-tier artificial tactile peripheral nervous system with neuromorphic signal processing capability, which can extract geometric features of the measured object. The system adopts the design method of overlapping structure so that each receptive field can overlapped each other, thereby improving the spatial resolution. This result is consistent with the theoretical simulations of previous studies [98], and provides an effective way for artificial skin to lead to the special ability of human fingertips to recognize fine textures. This work successfully emulates the neural structure through the combination of TENG and integrated circuits, which provides a good idea for the development of neural systems in the future.

## 4. Summary and Perspectives

In this review, the research progress of TENGs as active tactile stimulators for multifunctional sensing and artificial synapses is systematically reviewed. The applications of TENGs as active tactile stimulators for pressure sensing, temperature sensing, proximity sensing and object recognition are summarized in detail. In addition, the research progress of TENGs as active tactile stimulators for artificial synapses is emphatically analyzed, which is mainly reflected in the electrolyte-gate synaptic transistors, optoelectronic synaptic transistors, floating-gate synaptic transistors, reduced graphene oxides-based artificial neuromorphic synaptic, and integrated circuit-based artificial synapse and nervous systems. Based on the contact electrification and electrostatic induction, the ability of various synaptic devices to perceive external stimuli is described, and their synaptic plasticity such as EPSC, PPF, and STM to LTM is reviewed. For multifunctional sensing and artificial synapses based on TENGs, the triboelectric potential generated by TENGs plays two key roles. One role is that the input information of the external stimulus is directly related to the output signal, which realizes active sensing and improves the interference problem in the process of signal processing. The other role is that the triboelectric potential can be used as the gate power source of the synaptic devices, thus reducing the total power consumption of the devices.

Although gratifying development of TENGs as active tactile stimulators for multifunctional sensing and artificial synapses has been demonstrated, many challenges still need to be solved and further research needs to be carried out.

Active tactile stimulators based on TENGs not only require high performance such as high sensitivity, high resolution, wide detection range, fast response, and good linearity, but also require functional characteristics such as stretchability, transparency, and self-healing. Although the tactile stimulators based on TENGs show ultra-high sensitivity under low pressure, the sensitivity decreases with the increase of pressure, which greatly limits measurement range. This is mainly because the output signal of the device is closely related to the contact area of the friction layer. Under low pressure, the contact area gradually increases as the pressure increases. When the pressure increases to a certain value, the contact area gradually saturates, which reduces the sensitivity. In addition, the characteristics such as stretchability, transparency, and self-healing are essential elements for the practical application of TENGs-based active tactile stimulators to complex environments. Therefore, the structure optimization of devices and the development of new materials to obtain superior comprehensive performance are still the research focus in the future.Facts have proven that the active tactile stimulators based on TENGs can realize multifunctional sensing, including pressure and temperature sensing, pressure, and proximity sensing, pressure, and object recognition. However, the currently reported TENGs-based active tactile stimulators are not fully capable of simultaneously sensing more external stimuli similar to human skin in practical applications, since human skin has a variety of sensory receptors, such as mechanoreceptors, pain receptors, cold receptors, warm receptors. In addition to the research on structure and materials of the devices, integrating the TENG with other sensing mechanisms into a sensing unit to respond to multiple stimuli in a complex environment is also research topic in the future.The human brain is composed of 10^11^ neurons and 10^15^ synapses, which can handle various complex tasks with a small volume occupation and an ultra-low power consumption. However, current research mainly focuses on the single synaptic device based on TENG that can mimic part of the synaptic behavior, but there are few reports on the array device that mimic the parallel operation of the brain. This is because the high-density integration and miniaturization of the array device, which leads to huge challenges in micro/nano manufacturing technology. Therefore, how to ensure the consistency of the performance of each unit of the array device is one of the main problems. Another thorny issue is that high-density integrated array device is prone to signal interference between units, which affect the output signal of the array device. In future research, it is necessary to optimize the preparation process and design an effective anti-interference scheme to achieve large-scale parallel work and high plasticity of the array devices.The effective combination of the triboelectric potential and synaptic transistor not only establishes the active sensing mechanism between external stimuli and the device, but also effectively reduces the total power consumption of the device. Although the device has successfully mimicked the function of receptors and synapses in biological systems, but there is still a long way to go to realize a complete bionic perception system, including signal perception, signal transmission, signal processing and signal feedback. Therefore, it is necessary to design a variety of different stimulators by using different mechanisms or structures to mimic tactile, auditory, visual, and other sensory functions, and make them work synergistically to update synaptic weights to achieve interactive neuromorphic computation. In addition, it will be an effective strategy to integrate the device with the signal processing circuit and build a complete bionic sensing closed-loop system driven by new software or algorithm.

In conclusion, although the development of active tactile stimulators based on TENGs still faces some challenges and problems, we believe that through the interdisciplinary integration of microelectronics, physics, materials science, computer science, etc., active tactile stimulators based on TENGs will have bright prospects and broad application potential.

## Figures and Tables

**Figure 1 sensors-22-00975-f001:**
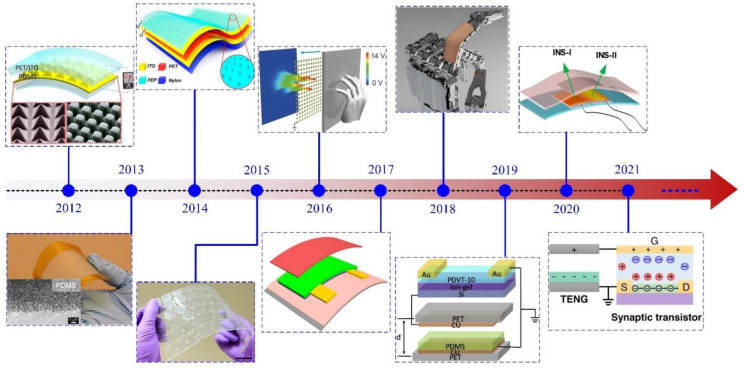
Brief timeline of tactile stimulator based on triboelectric nanogenerators. Reproduced with permission [41]. Copyright 2012, American Chemical Society. Reproduced with permission [21]. Copyright 2013, American Chemical Society. Reproduced with permission [22]. Copyright 2014, American Chemical Society. Reproduced with permission [33]. Copyright 2015, American Chemical Society. Reproduced with permission [43]. Copyright 2016, John Wiley and Sons. Reproduced with permission [44]. Copyright 2017, American Chemical Society. Reproduced with permission [48]. Copyright 2018, John Wiley and Sons. Reproduced with permission [49]. Copyright 2019, Elsevier. Reproduced with permission [50]. Copyright 2020, American Chemical Society. Reproduced with permission [51]. Copyright 2021, Springer Nature.

**Figure 2 sensors-22-00975-f002:**
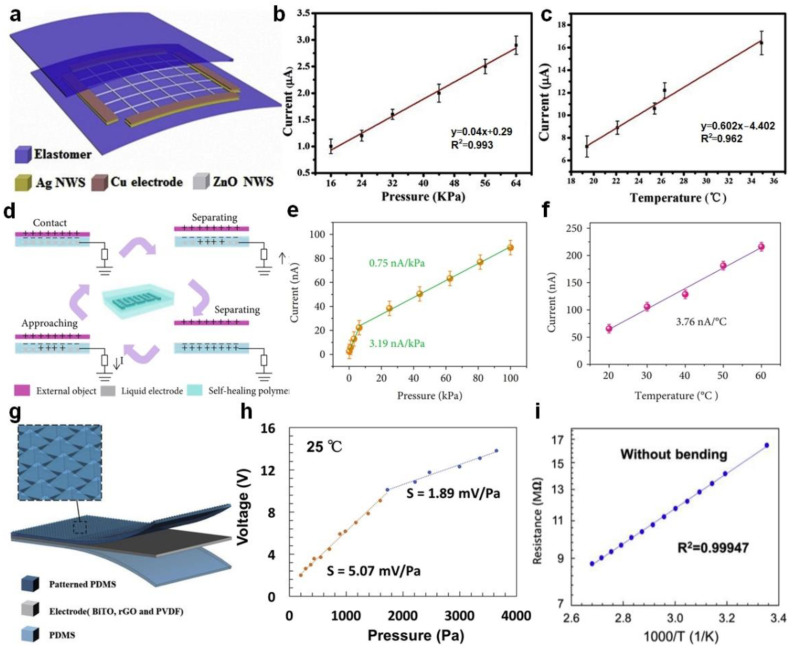
Pressure and temperature sensing. (**a**) Schematic diagram of the tactile stimulator. (**b**) Functional relationship between external pressure and the current. (**c**) Functional relationship between temperature and the current. Reproduced with permission [54]. Copyright 2019, Elsevier. (**d**) The working principle of the self-healing electronic skin. (**e**) Pressure response curve of the self-healing electronic skin. (**f**) Temperature response curve of the self-healing electronic skin. Reproduced with permission [55]. Copyright 2021, American Association for the Advancement of Science. (**g**) Schematic illustration of the tactile electronic skin. (**h**) Functional relationship between external pressure and output voltage at room temperature. (**i**) Functional relationship between reciprocal temperature and electrode resistance without deformation. Reproduced with permission [56]. Copyright 2020, Elsevier.

**Figure 3 sensors-22-00975-f003:**
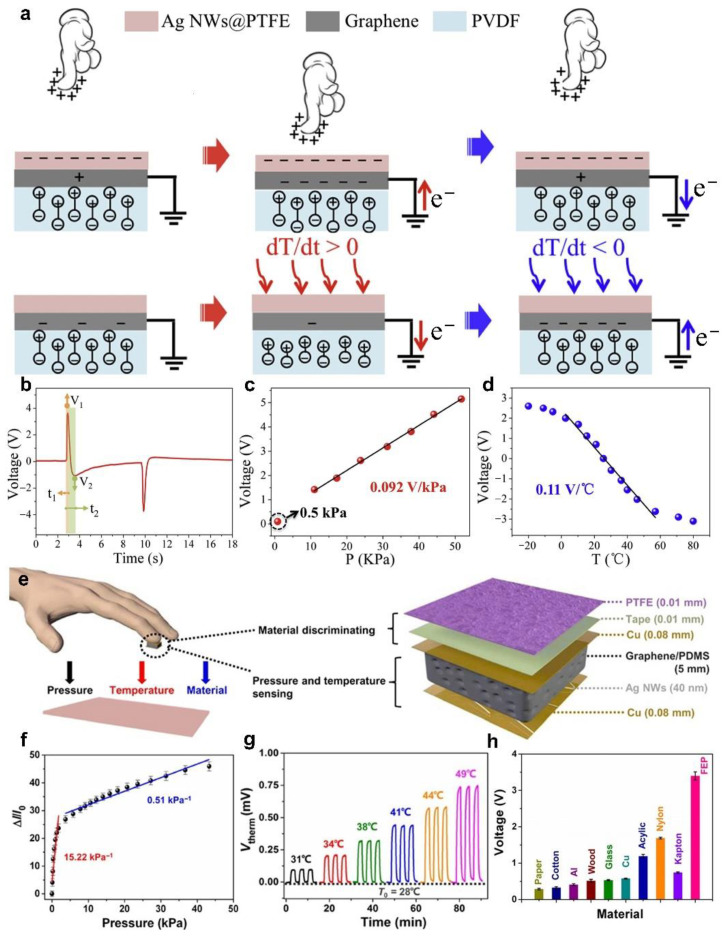
Pressure and temperature sensing. (**a**) Schematic diagram of pressure and temperature sensing mechanism based on multi-effect coupling. (**b**) The voltage response curve under the coupling of pressure and temperature stimuli. (**c**) Functional relationship between external pressure and output voltage. (**d**) Functional relationship between temperature and output voltage. Reproduced with permission [57]. Copyright 2019, Elsevier. (**e**) Schematic diagram of the multifunctional self-powered tactile stimulator. (**f**) Pressure response curve of the device. (**g**) Temperature response curve of the device. (**h**) The voltage peak amplitude generated by contact between the device and different materials. Reproduced with permission [58]. Copyright 2020, American Association for the Advancement of Science.

**Figure 4 sensors-22-00975-f004:**
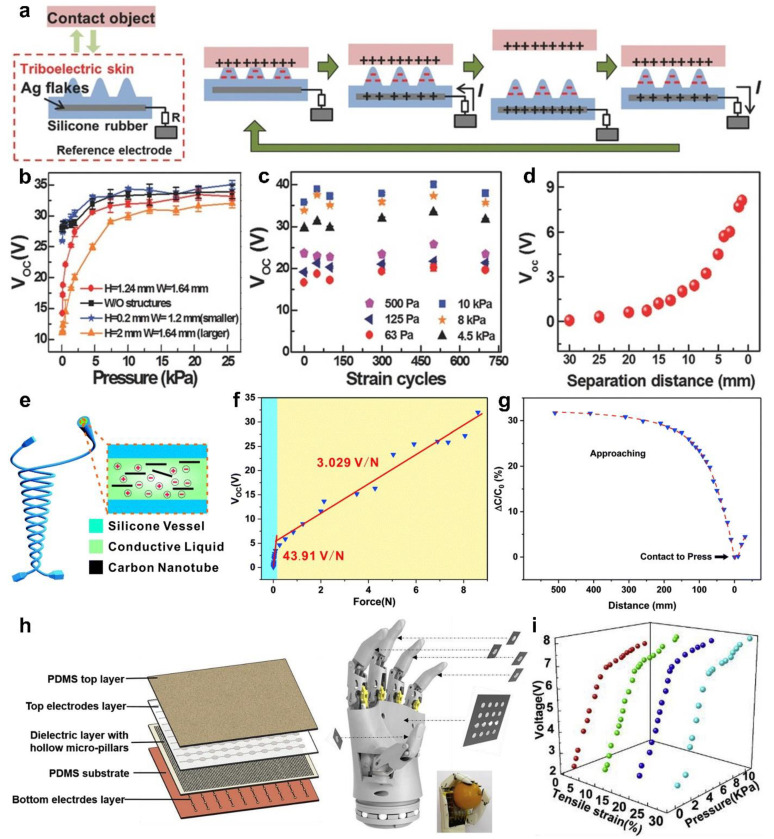
Pressure and proximity sensing. (**a**) Schematic illustration of the working mechanism of the triboelectric skins. (**b**) Functional relationship between pressure and the voltage of devices with different structures. (**c**) The variation characteristics of the voltage under different pressure and strain cycles. (**d**) Functional relationship between separation distance and the voltage. Reproduced with permission [59]. Copyright 2018, John Wiley and Sons. (**e**) Schematic diagram of the liquid-polymer tubular TENG. (**f**). Functional relationship between force and open-circuit voltage. (**g**). Functional relationship between distance and capacitance. Reproduced with permission [60]. Copyright 2019, Royal Society of Chemistry. (**h**) Schematic diagram of the stretchable dual-mode sensor array for a bionic robot. (**i**) Functional relationship between pressure and open-circuit voltage under different tensile strain. Reproduced with permission [61]. Copyright 2019, Elsevier.

**Figure 5 sensors-22-00975-f005:**
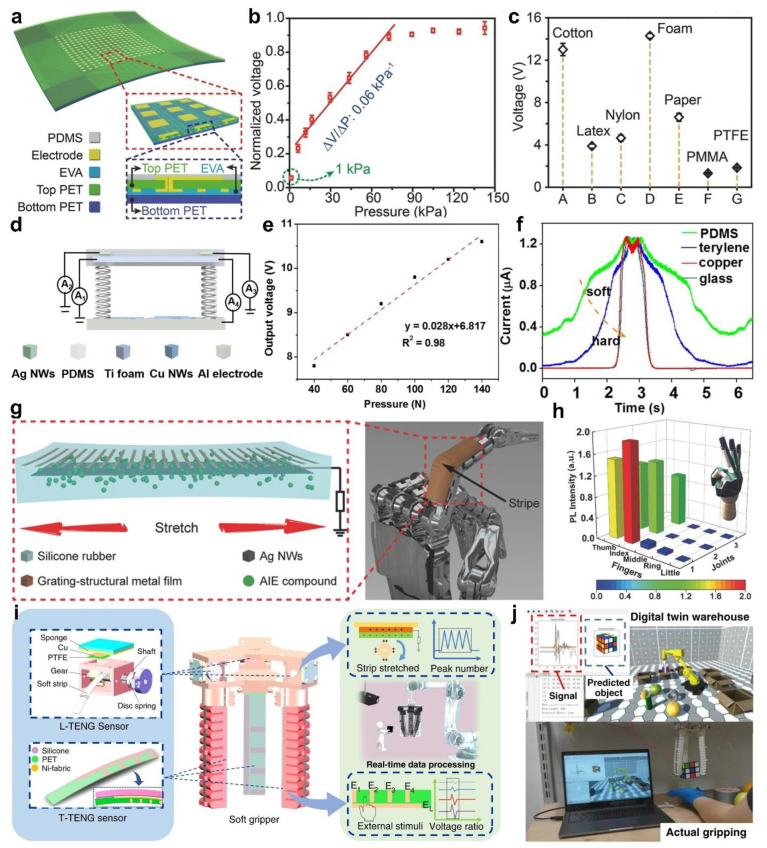
Pressure and object recognition. (**a**) Schematic diagram of the triboelectric sensor matrix. (**b**) Functional relationship between pressure and the output voltage. (**c**) The output voltage is generated by contact between the device and different materials. Reproduced with permission [43]. Copyright 2019, John Wiley and Sons. (**d**) Schematic diagram of the tactile stimulator. (**e**) Functional relationship between pressure and the output voltage. (**f**) The current change is generated by contact between the device and different materials. Reproduced with permission [62]. Copyright 2017, American Chemical Society. (**g**) Schematic diagram of the triboelectric-photonic smart skin for a robotic finger. (**h**) The 3D normalized photoluminescence intensity map of a hand gesture “OK”. Reproduced with permission [48]. Copyright 2018, John Wiley and Sons. (**i**) Construction drawing of the TENG for soft gripper. (**j**) Actual gripping and virtual demonstration of object recognition. Reproduced with permission [63]. Copyright 2020, Springer Nature.

**Figure 6 sensors-22-00975-f006:**
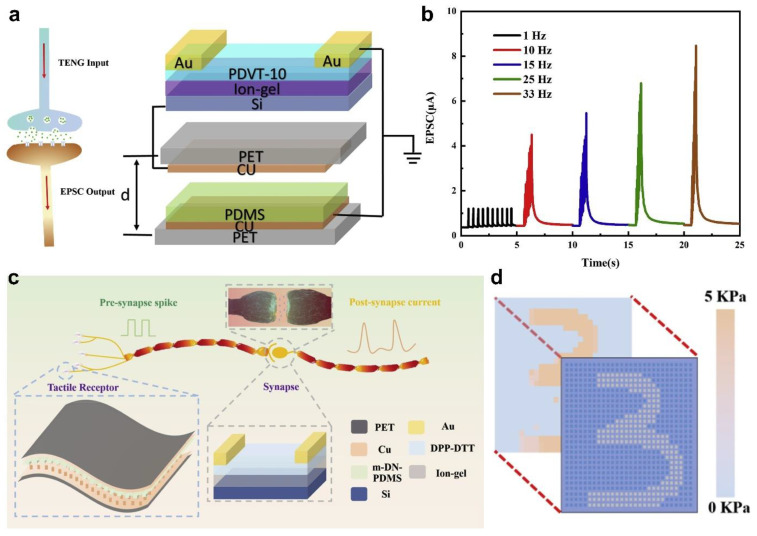
Electrolyte-gate synaptic transistors based on organic semiconductor material. (**a**) Schematic diagram of the SPST. (**b**) EPSC of the SPST at different touch frequencies. Reproduced with permission [49]. Copyright 2019, Elsevier. (**c**) Schematic diagram of the artificial sensory memory system. (**d**) Diagrams of the handwritten digit number and the tactile mapping. Reproduced with permission [70]. Copyright 2020, Elsevier.

**Figure 7 sensors-22-00975-f007:**
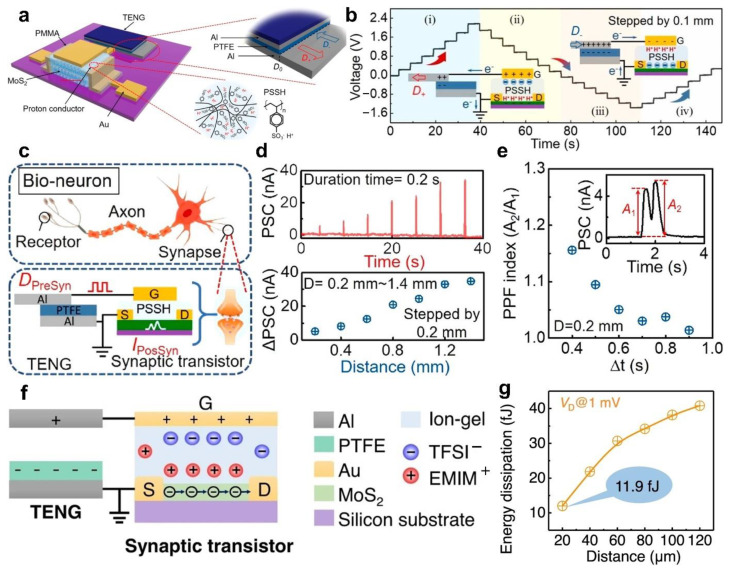
Electrolyte-gate synaptic transistors based on two-dimensional semiconductor materials. (**a**) Schematic diagram of the triboiontronic synaptic transistor. (**b**) The output voltage of the TENG under the action of a displacement pulse. (**c**) Schematic diagram of a bio-neuron and the triboiontronic synaptic transistor. (**d**) The PSCs of the transistor under the action of a displacement pulse. (**e**) Functional relationship between time interval (Δt) and PPF index. Reproduced with permission [85]. Copyright 2020, American Chemical Society. (**f**) Schematic diagram of triboelectrification-activated artificial afferent neuron. (**g**) Functional relationship between displacement and energy dissipation. Reproduced with permission [51]. Copyright 2021, Springer Nature.

**Figure 8 sensors-22-00975-f008:**
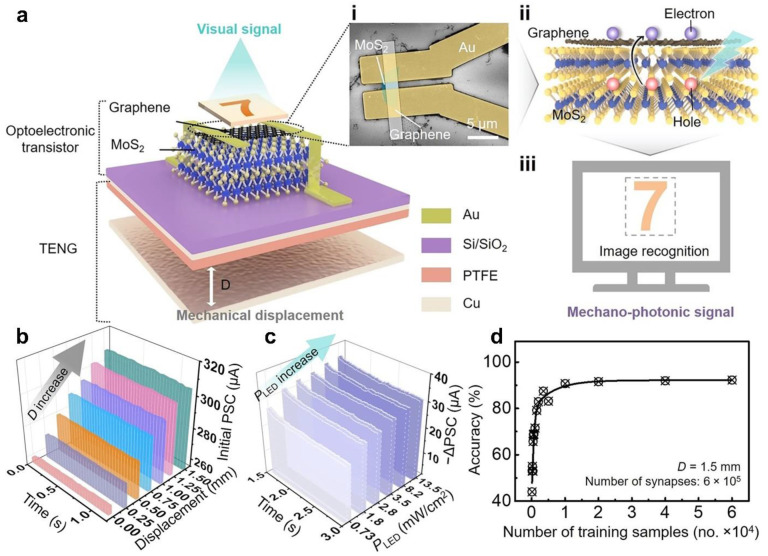
Mechano-photonic artificial synapse. (**a**) Schematic diagram of the mechano-photonic synapse device based on graphene/MoS_2_ heterostructure. (**i**) Scanning electron microscope image of the top of the synapse device. (**ii**) Schematic diagram charge transfer/exchange for the graphene/MoS_2_ heterostructure. (**iii**) Image recognition of the synapse device in response to mechano-photonic signal. (**b**) Functional relationship between displacement and real-time initial PSC in the dark. (**c**) Functional relationship between light intensity and real-time −ΔPSC. (**d**) Functional relationship between numbers of training samples and recognition accuracy. Reproduced with permission [91]. Copyright 2021, American Association for the Advancement of Science.

**Figure 9 sensors-22-00975-f009:**
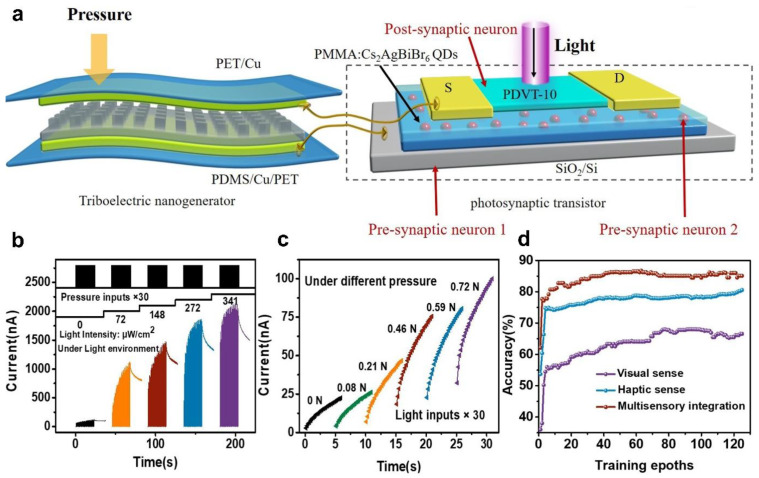
Artificial multisensory integration nervous system based on TENG and photosynaptic transistor. (**a**) Schematic diagram of the artificial multisensory integration nervous system. (**b**) The variation characteristics of the channel current under 30 continuously touch pulses and different light intensity. (**c**) The variation characteristics of the channel current under 30 continuously optical pluses and different pressure. (**d**) Functional relationship between numbers of training epochs and recognition accuracy. Reproduced with permission [93]. Copyright 2021, Elsevier.

**Figure 10 sensors-22-00975-f010:**
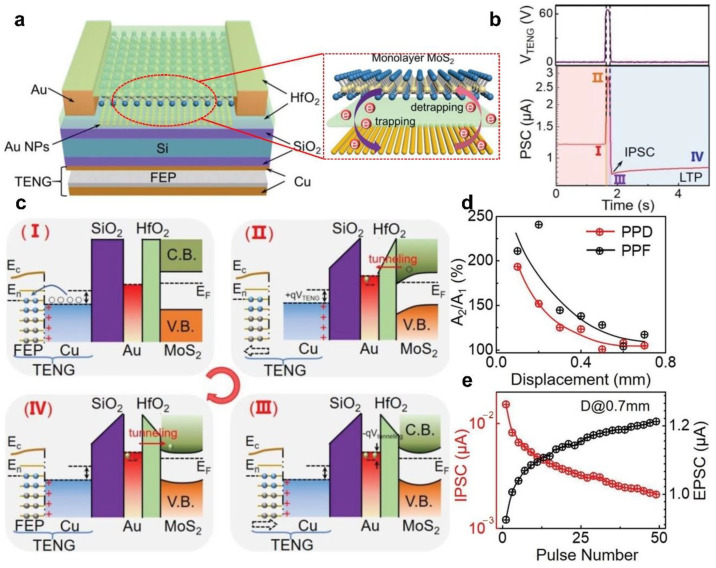
Mechanoplastic tribotronic floating-gated synaptic transistor. (**a**) Schematic diagram of the floating-gated synaptic transistor. (**b**) The PSC of the transistor under the action of a displacement pulse. (**c**) Schematic diagram of the working mechanism of the transistor, (**I**–**IV**) are four different working states, respectively. (**d**) Functional relationship between displacement and paired pulse depression/facilitation index. (**e**) Functional relationship between pulse number and EPSC/IPSC. Reproduced with permission [94]. Copyright 2020, John Wiley and Sons.

**Figure 11 sensors-22-00975-f011:**
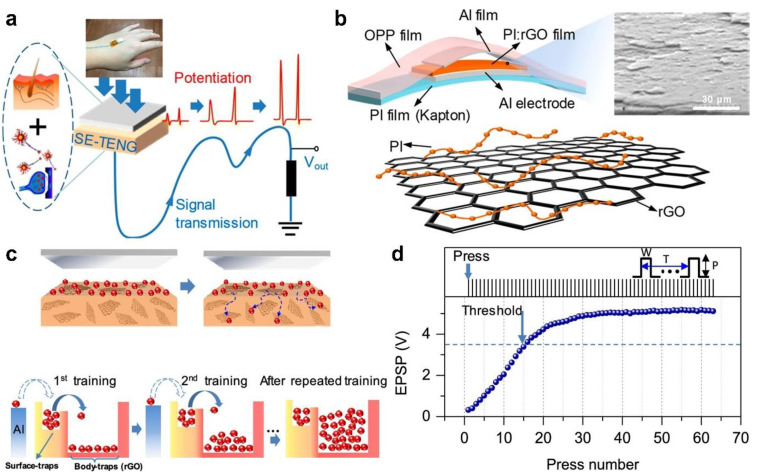
Reduced graphene oxides-based artificial synapse. (**a**) Conceptual diagram of the artificial synapse. (**b**) Schematic structure of the artificial synapse. (**c**) Schematic diagram of the triboelectric electrons transfer process. (**d**) Functional relationship between press number and output voltage of the artificial synaptic device. Reproduced with permission [50]. Copyright 2019, American Chemical Society.

**Figure 12 sensors-22-00975-f012:**
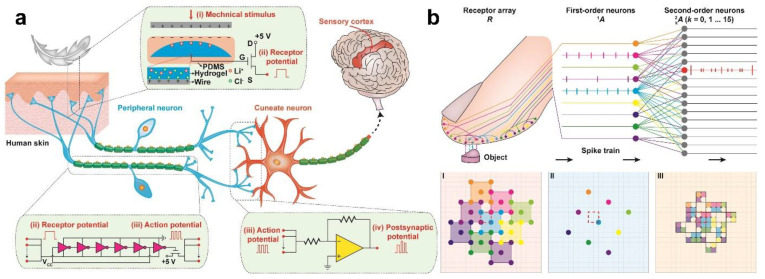
Integrated circuit-based artificial synapse and peripheral nervous system. (**a**) Schematic diagram of a bioinspired artificial tactile peripheral nervous system. (**b**) Spatial recognition of the two-tier artificial tactile peripheral nervous system. (**I**) Layout of the mechanoreceptor. (**II**) The mapping of the mechanoreceptors to artificial peripheral neurons. (**III**) The synaptic structures. Reproduced with permission [97]. Copyright 2021, Elsevier.

**Table 1 sensors-22-00975-t001:** Comparisons of active tactile stimulators based on TENG for pressure and temperature sensing.

Reference	Size	Triboelectric Materials	Detection Signal	Sensitivity	Detection Range
[54]	40 × 40 mm	(OMMT-CTAB)	Pressure: short-circuit current	0.04 μA/kPa	16~64 kPa
			Temperature: short-circuit current	0.59 μA/°C	19.4~34.9 °C
[55]	20 × 20 mm	Self-healing polymer	Pressure: output current	3.55 nA/kPa	0.12~100 kPa
			Temperature: output current	3.76 nA/°C	20~60 °C
[56]	15 × 10 mm	PDMS	Pressure: output voltage	5.07 mV/Pa	200~3650 Pa
			Temperature: electrode resistance	β_25/100_ ≈ 1024 K	25~100 °C
[57]	50 × 50 mm	Ag NWs @PTFE	Pressure: output voltage	0.092 V/kPa	10.5~52 kPa
			Temperature: output voltage	0.11 V/°C	10~45 °C
[58]	10 × 10 mm	PTFE	Pressure: output current	15.22 /kPa	0~40 kPa
			Temperature: output voltage	35.2 μV/K	25~60 °C

## Data Availability

Not applicable.

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
