# Peer review of "Triboelectric Nanogenerators as Active Tactile Stimulators for Multifunctional Sensing and Artificial Synapses"

_sensors, 2022, doi:10.3390/s22030975_

Round 1

Reviewer 1 Report

In this review paper, the authors give a thorough review of the recent progress of triboelectric nanogenerators (TENGs) based active tactile stimulators for multifunctional sensing and artificial synapses. The manuscript is well organized and well written. And I think this review is of great interest to the relevant field such as microelectronics, physics, and materials science. Therefore, I would recommend the publication of the work in Sensors once some minor issues are addressed.

  1. Page 6, line 201. Could the authors clarify that if the sensitivity for pressure sensing of 15.22 kPa-1 is measured from its TENG output signal or the resistance of spongy graphene/PDMS composite? Moreover, Figure 3f isn’t mentioned in this manuscript.
  2. Page 6, line 203. “the temperature range of 0-22K” doesn’t make sense. I think it should be the temperature difference here. Also, it would be better if the actual temperature range is provided.
  3. Page 14, line 446. “1~100 fj” should be ”1~100 fJ”. Similar minor error/typo appears several times throughout this manuscript, such as 0.1s, 2.8 μ A. The authors should check and correct them.

Reviewer 2 Report

In the work, the authors have summarized a number of very interesting research works concerning using TENGs as active tactile stimulators for multifunctional sensing and artificial synapses. At the end of the manuscript, a short conclusion has been made, and also some perspertives which indicate several challenging points in the above-mentioned domain. Certainly, the topics are quite interesting because they are currently quite hot research themes.

The manuscript has also been well written in terms of readability. 

However, I think this manuscript seems a summary of certain published papers (abstracts + representative figures), instead of a “critical” review. This latter would be more interesting for readers and would inspire them to resolve the related problems in the domain. This is one significant shortcoming of this work.

In addition, the figure 1 contains too many figures to be clearly and easily read.

Reviewer 3 Report

The article reviewed triboelectric nanogenerators. Tactile sensors and synaptic transistors using the triboelectric nanogenerators are explained. Also, the results of the references are explained. The problem of this paper is lack of the discussion. The results of the references are only explained and there is no useful discussion. Therefore, I did not recommend an editor to accept this article and show some problems.

  • I think that a section to explain TENG is necessary. The article requires the fundamental knowledge for TENS to readers. 
  • Some articles are explained in detail. Is there some reason why the articles are selected?
  • This article only shows and explained the figures of the references. I do not understand the relationships of the references. For example, adding a table for specification (Sensitivity, size, and weight) will be useful for readers to understand the advantages and disadvantages. Please check other review articles to write a useful and understandable article.
    • Hwang et al., Non-planar PDMS microfluidic channels and actuators: A review, Lab on a Chip.

    • Pacchierotti et al., Wearable Haptic Systems for the Fingertip and the Hand: Taxonomy, Review, and Perspectives, IEEE ToH. 
  • There should be many discussion provided by the authors. Current article is only summarized of some articles. It would be better for readers to read original articles. Please reconsider entire structure of this paper. 

Round 2

Reviewer 2 Report

Thank very much the authors for the revision of the manuscript and the corresponding improvements made mainly in the first part (introduction) and the last part (conclusion).  However, regarding the most important main parts (2 and 3), few changes have been made. Personally, it would be preferable to significantly reduce the citation of the results and add more corresponding discussion and analyses.  

Reviewer 3 Report

The authors conducted minor revision according to the reviewer's comments,
This article is still the summary of the references and lacks author's unique view point.

I did not agree with acceptance for this article.
